# Live-shaping of hydrogel thin films with light

Matias Paatelainen, Henning Meteling, Alex Berdin  & Arri Priimagi  ✉

Light-responsive hydrogels are attractive materials for mimicking dynamic microstructures in nature, providing platform for tunable devices in photonics, sensing, and biomedicine. However, these systems often suffer from limited spatial resolution and slow response times, restricting their utility in high-speed or high-precision applications. Here, we present a light-responsive hydrogel thin film platform capable of rapid, reconfigurable surface modulation with sub-micron spatial resolution and actuation frequencies up to 2 Hz. The system leverages photoswitchable host–guest interactions to induce reversible contraction–expansion in response to patterned illumination. Dual-wavelength control enables the generation of dynamic, migrating surface features capable of transporting micro-objects in real time. The approach is further extended to free-standing films, demonstrating functionality in laser beam steering. Additionally, surface patterns can be stabilized by drying and erased by humidity, offering a route to rewritable sensor tags.

In nature, various microscopic surface structures serve purposes such as wetting control, adhesion, mate attraction, and camouflage[1–3]. Some structures are irregular, like the water-repellent surfaces of rose petals and lotus leaves covered with micropillars[4,5]. Others require high precision, such as photonic crystals on butterfly wings and peacock feathers, creating structural colors[6,7]. While these examples can be considered passive, nature has also engineered sophisticated, dynamic variants, like the skin of cephalopods and chameleons. Cephalopods control dynamic color pigment expression, whereas chameleons have adjustable photonic crystals on their skin to create structural colors, providing camouflage and communication tools[8,9].

Inspired by these natural examples, artificial dynamic surfaces have been envisioned for many applications, such as wetting control[10,11], sensing[12,13], active photonic components[14], and dynamic cell culture platforms[15,16]. Soft materials, particularly hydrogels, are beneficial for mimicking living functions, due to their tunable chemistry, tissue-like physical properties, and good biocompatibility[17,18]. Furthermore, they can be controlled externally by various stimuli, e.g., temperature[19,20], light[21,22], pH[23,24], electric and magnetic fields[25,26], and different analytes[12,27].

Light is a particularly interesting stimulus for systems requiring high precision and non-invasive control, as it can be modulated accurately in terms of dosage, spatial distribution, and polarization. For example, light-responsive hydrogels have been demonstrated for remotely triggered drug delivery[28], soft actuators[29], and mechano-active cell platforms[30]. One frequently studied mechanism for light-

responsive hydrogels revolves around inclusion complexes. These systems can, in general, enable dynamic crosslinking, thereby significantly improving mechanical properties and self-healing ability[31–34]. When such inclusion complexes are made light-responsive through the incorporation of a photoswitchable guest, their properties can be modulated remotely with high spatiotemporal control[35,36]. An extensively studied example is provided by inclusion complexes between azobenzene photoswitches and cyclodextrins. In these systems, host–guest complexation can selectively favor one isomer over the other, resulting in pronounced hydrophilicity contrast between the two states.

The light-controllable host–guest complexation has been widely implemented, ranging from control over supramolecular gelation[37] and self-assembly[38] to macroscopic deformations in soft robotic systems[39], microactuators[40], and artificial muscles[41,42]. Recently, this principle was exploited to create reconfigurable micropatterns on hydrogel films by integrating azobenzene:α-cyclodextrin complexes[43]. Photoisomerization of azobenzene allows for localized control over film swelling and contraction, yielding surface patterns with spatial resolution down to 40 μm in less than a minute. Importantly, these patterns can be erased and rewritten by photoisomerization of azobenzene, enabling fully reversible architecture that acts as a versatile micromolding platform for complex surface features. Despite the recent advancements, light-responsive hydrogels suffer from relatively slow actuation due to diffusion-controlled contraction–expansion, which strongly scales with dimensions. This typically limits the

Smart Photonic Materials, Faculty of Engineering and Natural Sciences, Tampere University, Tampere, Finland. ✉e-mail: arri.priimagi@tuni.fi

actuation timescales somewhere between tens of seconds to minutes[30,43,44]. Thus, the bar is high for applications like body action recreation, such as respiration, heartbeat, and dynamic camouflage. Similarly, the resolution of spatial deformation does not quite reach the dimensions comparable to the wavelength of visible light but falls somewhere around tens or hundreds of micrometers and therefore poses limitations to photonics applications[43,45].

Here, we present a method for fast, reconfigurable, and dynamic shaping of hydrogel thin films with light. Surface-attached hydrogel films can be exposed to arbitrary illumination patterns, which are translated into controlled surface features. These features can be produced in sub-second timescale, with sub-micron spatial resolution and full reconfigurability. Due to the rapid response of the system, contraction–expansion cycles can operate at 2 Hz frequency, reaching the resting heart rate of adults. Moreover, dual illumination with visible and UV light enables dynamic surface structures that evolve and migrate on the surface in real time, even transporting small objects across the surface. The fast dynamics can also be adapted to free-standing hydrogel films, where we demonstrate chameleon-skin-inspired tunable coloration and laser beam steering. In addition, by drying the hydrogel in ambient conditions, the inscribed surface structures can be locked in and later erased at will with increased humidity, suggesting promising platform for, e.g., reconfigurable sensor tags.

## Results

### Material design principle

The swelling dynamics of hydrogels are increasingly diffusion-controlled as the volume-to-surface ratio rises, leading to a slower response to external stimuli. Furthermore, free-standing hydrogels typically undergo isotropic swelling upon stimulation, in contrast to constrained systems where mechanical restrictions can result in anisotropic movements, as depicted in Fig. 1a. A fabrication method with accurate control over the hydrogel dimensions is therefore essential for fast and precise shaping, which we realize with materials design shown in Fig. 1b. The first fabrication step involves thermal free-radical polymerization of the hydrophilic monomer N-isopropylacrylamide (NIPAm) together with in-house synthesized photoactive crosslinker 4-acrylamidobenzophenone (BP) and photoswitchable 4-acrylamidoazobenzene (AZO) (I, SI, Fig. S1–S5). The intermediate oligomeric stage allows spin-coating of uniform thin films (II), which can be crosslinked by activating the BP groups with UV light (III). This provides both crosslinking and surface anchoring, resulting in surface-attached poly(NIPAm-co-AZO-co-BP) hydrogel (PNIPAm-AZO) thin films with uniform thickness.

After the hydrogel fabrication, the host molecule α-cyclodextrin (αCD) is introduced into the system by immersing the hydrogel film in aqueous solution (IV). In an aqueous environment, the hydrophobic E-AZO molecules are encapsulated by αCD, whose hydrophilic exterior enhances the swelling of the hydrogel[39]. For uncrosslinked

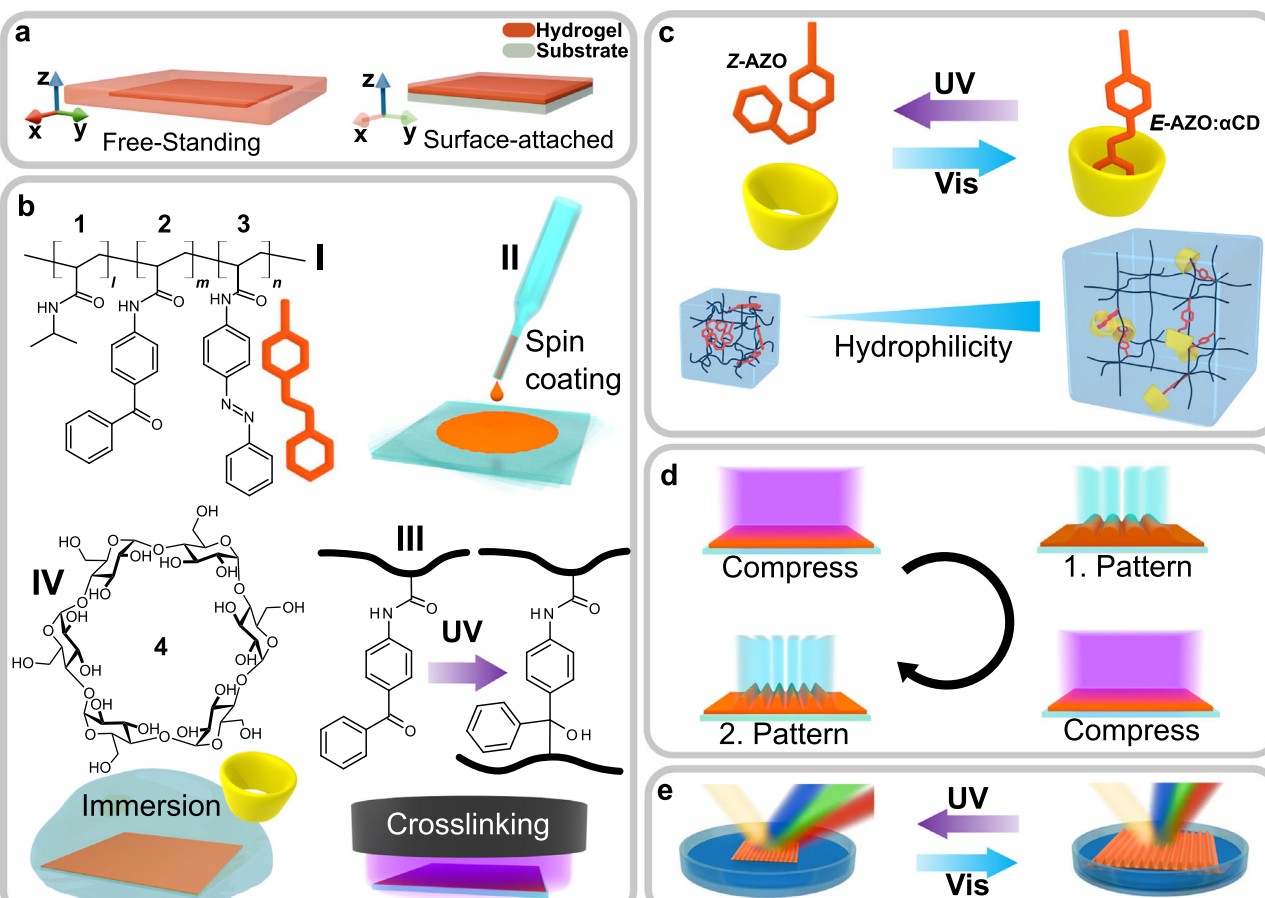

**Fig. 1 | Reversible surface structuring and morphing of photoresponsive host–guest hydrogel films. a** Comparison of isotropic xyz-swelling in free-standing films versus anisotropic z-directional expansion in surface-attached films. **b (I)** Oligomer composition: (1) N-isopropylacrylamide, (2) 4-acrylamidobenzophenone (BP) photocrosslinker, and (3) 4-acrylamidoazobenzene (AZO) moiety. **(II–IV)** Film fabrication via spin-coating, photocrosslinking, and subsequent functionalization through immersion in aqueous (4) α-cyclodextrin (αCD) solution. **c** The functioning principle: The E-AZO:αCD inclusion complex increases network hydrophilicity, driving expansion, whereas Z-AZO dissociation reduces it. **d** Reconfigurable photopatterning: UV-induced compression followed by spatially modulated visible light exposure enables spatially controlled complexation and expansion. Features are erasable and rewritable through subsequent light cycles. **e** Phototunable diffractive structures demonstrated via contraction–expansion of surface relief gratings (SRG) on free-standing films.

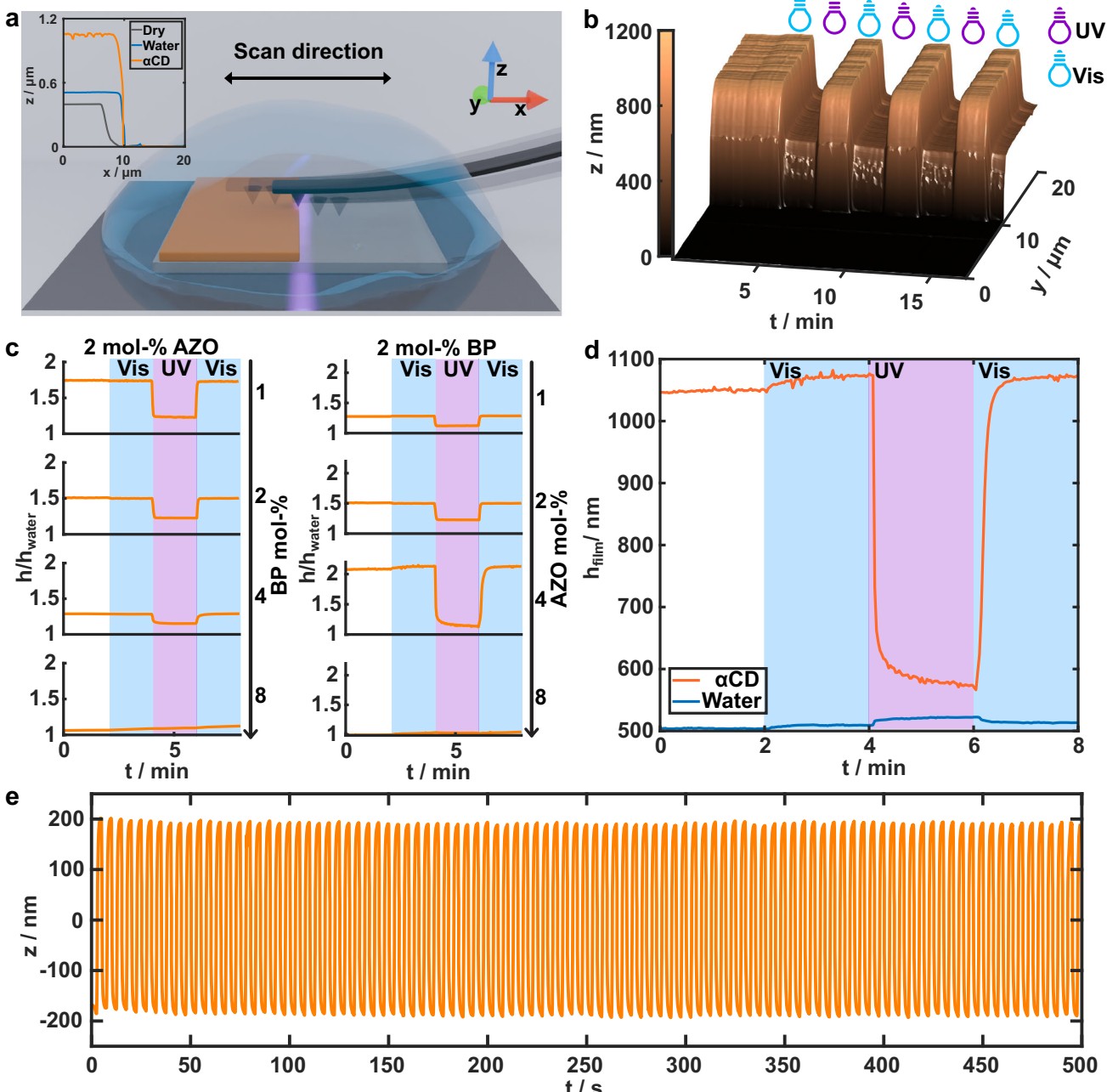

**Fig. 2 | Light-responsive contraction–expansion of surface-attached hydrogel thin films. a** Schematic of in-situ AFM thickness measurements during simultaneous irradiation. Inset: typical film step-profiles for a hydrogel (4 mol-% AZO, 2 mol-% BP) in air, water, and αCD solution. **b** Time-resolved step-profiles illustrating film contraction–expansion during photoswitching. **c** Contraction–expansion response in 100 mg mL⁻¹ αCD as a function of BP (left) and AZO (right) concentrations; y-axis represents thickness relative to the initial state in water (h/h_water).

**d** Absolute film thickness modulation of hydrogel (4 mol-% AZO, 2 mol-% BP) in αCD solution (100 mg mL⁻¹) versus pure water. **e** Contraction–expansion of hydrogel film (4 mol-% AZO, 2 mol-% BP) for 100 cycles, monitored via digital holographic microscopy. Irradiation parameters: **b–d**: 30 mW cm⁻² at 365 nm (UV) and 70 mW cm⁻² at 490 nm (Vis) **e**: 100 mW cm⁻² at 365 nm (UV) and 170 mW cm⁻² at 490 nm (Vis) with 2.5 s exposure per wavelength.

poly(NIPAm-co-AZO) in aqueous solution, the selective binding and association constant was determined as $K_{[E-AZO:\alpha CD]} = 2\times10^3$ M⁻¹ from UV-Vis and NMR titration (Figs. S6 and S7), which is consistent with a previously reported value for side-chain azobenzene[46]. On the contrary, Z-AZO cannot fit inside the αCD cavity due to steric effects, causing the complex to dissociate and the overall hydrophilicity to decrease (Fig. S8). The E−Z isomerization of AZO can be efficiently driven with UV and visible light in the hydrogel under aqueous conditions (Fig. S9), enabling reversible control over the host–guest complexation and, thus, the swelling of the hydrogel (Fig. 1c). As will be

demonstrated, this mechanism can be used to create arbitrary surface features on surface-attached films, as well as to dynamically modulate pre-inscribed surface features on free-standing films (Fig. 1d, e).

## Light-Induced Contraction and Expansion

To characterize the light-responsive contraction and expansion of the surface-confined hydrogel films, we measured the film thicknesses for samples with different AZO and BP fractions in water with atomic force microscope (AFM) under simultaneous illumination (Fig. 2a). The samples were covered with 100 mg mL⁻¹ αCD water solution, which

proved to be a sufficient and practical concentration for effective swelling and was used in following experiments unless otherwise described (Fig. S10). Imaging was performed by scanning across a step profile between the substrate and the hydrogel film for thickness determination (Fig. 2a, inset). The profile was repeatedly recorded along the same line over the sample to monitor the dynamics of film contraction and expansion under UV and visible illumination, as shown in Fig. 2b. The light-induced contraction and expansion of the AZO and BP concentration series, relative to the film thickness in water, are shown in Fig. 2c. In the BP series, the AZO concentration was fixed at 2 mol-%. Similarly, the BP concentration was kept constant at 2 mol-% in the AZO series. The intensities in all AFM measurements were estimated as 30 mW cm$^{-2}$ for 365 nm and 70 mW cm$^{-2}$ for 490 nm (see Methods).

As expected, increasing the BP concentration led to a reduction in maximum expansion, which can be attributed to a more restricted network structure resulting from higher crosslinking density[47]. Increasing the AZO concentration initially leads to a stronger response. This is likely due to higher overall hydrophilicity contrast between the $E$-AZO:αCD complex and free $Z$-AZO, as the fraction of light-responsive moieties increases. Interestingly, the complexation- and decomplexation-induced contraction–expansion ceases as the AZO concentration is further increased from 4 to 8 mol-%. We hypothesize that this phenomenon is related to the lower critical solution temperature (LCST) behavior of PNIPAm, which is influenced by the presence of additional hydrophobic and hydrophilic moieties in the polymer[48,49]. In the currently studied hydrogel, both isomers of AZO are relatively hydrophobic compared to the NIPAm monomer, while the hydrophilicity of $E$-AZO:αCD complex is more similar to NIPAm than that of free $Z$-AZO. Thus, switching between the two states modulates the LCST of the system. To support this hypothesis, we studied the LCST behavior by measuring the turbidity of non-crosslinked PNIPAm-AZO in αCD solution as a function of temperature. It was observed that the LCST of a non-crosslinked oligomer with 4 mol-% of AZO and 2 mol-% of BP at a concentration of 2 mg mL$^{-1}$ and in the presence of 6 mg mL$^{-1}$ αCD shifts from 32 °C to 25 °C upon $E–Z$ isomerization of AZO and complex dissociation (Fig. S11). Consequently, at a given temperature range, the system can be switched between a hydrophilic and a hydrophobic state by photoisomerization of AZO. This indicates that the light-induced contraction and expansion is not solely driven by hydrophilicity contrast between $Z$-AZO and the $E$-AZO:αCD complex but is further amplified by a light-induced shift in the LCST. We also observed that the UV-Vis spectra of the $E$-AZO:αCD in solution remained unchanged upon heating to 37°C, suggesting that the complexation is not significantly affected by temperature within this range (Fig. S12). Therefore, at high AZO concentrations, the hydrogel remains collapsed regardless of αCD addition, likely due to a decrease in the LCST of the complexed hydrogel to below ambient temperature.

After varying both AZO and BP concentration, a composition containing 4 mol-% AZO and 2 mol-% BP exhibited the strongest light-response among the studied formulations. Hence, all subsequent experiments were conducted using this composition at room temperature, unless stated otherwise. The absolute thickness development of a hydrogel during photoswitching, both in the presence and absence of αCD, is shown in Fig. 2d. In the presence of αCD, a slight initial increase in sample thickness is observed upon exposure to visible light. This is likely due to pre-exposure to ambient and AFM illumination, which affects the initial $E$:$Z$ ratio of AZO. Subsequent UV irradiation triggers fast contraction, which is fully reversible upon visible light exposure, resulting in 87 % increase in sample thickness from the contracted to the expanded state. This simultaneously alters the mechanical properties, yielding an approximately 60% reduction in elastic modulus upon swelling (Fig. S13), consistent with previous reports on comparable hydrogel systems[50,51]. In addition, the dynamics and extent of contraction and expansion do not significantly change within dry film thickness range of 60-400 nm (Fig. S14). In the absence of αCD, the light-induced switching causes only a minor response in the opposite direction. This can be attributed to the dipolar nature of $Z$-AZO in contrast to the nonpolar $E$-AZO. However, the magnitude of the contraction and expansion without αCD is negligible compared to that observed in its presence (2 % $vs$. 87 %).

The robustness of the system was investigated using a digital holographic microscope (DHM; see Methods)[52]. A hydrogel film was subjected to periodic switching with 365 nm UV light (100 mW cm$^{-2}$) and 490 nm visible light (170 mW cm$^{-2}$) for 100 cycles, with a 5-second exposure per cycle, while monitoring the vertical displacement of the film surface (Fig. 2e). Film height oscillates with a consistent 400 nm amplitude, indicating stable and repeatable operation.

## From Light-Induced Contraction–Expansion to Dynamic Surface Structuring

To transition from simple contraction and expansion towards surface patterning of hydrogel films, we used a DHM equipped with a custom-made laser interference lithography setup[52]. A thin silver coating was sputtered onto the hydrogel to improve the imaging contrast in water solution (Fig. S15). We confirmed that the coating does not affect the light-responsive behavior by comparing the contraction–expansion dynamics on the plain hydrogel film and one with silver coating with AFM (Fig. S16). The coating does, however, cause slight additional attenuation when the sample is irradiated from above through the coating, rendering the light-response possibly slower compared to plain hydrogel. In addition, the agglomeration of the Ag layer under water leads to increased surface roughness. (Fig. S17).

The sequential patterning and erasure of two distinct surface relief gratings (SRGs) on the hydrogel surface using a 488 nm laser for interference irradiation and 365 nm UV LED for uniform illumination is demonstrated in Fig. 3a (200 mW cm$^{-1}$ and 0.2 s exposure time for both cases). Initially, the film is compressed with UV light (I) that causes $E–Z$-isomerization and breaks the $E$-AZO:αCD complexes. A laser interference pattern ($p$-polarized 488 nm) is then projected onto the film, giving rise to an SRG with 3 μm period via localized expansion in the regions of constructive interference (II). The surface structure can be subsequently completely erased with another UV exposure (III), and another SRG with a different period (5 μm in the example given) can be patterned on the same area (IV). The patterning–erasure cycles can be repeated, allowing fully reconfigurable surface structuring with pattern heights in the range of hundreds of nanometers, achieved within seconds.

An example of the surface structure formation and erasure dynamics is presented in Fig. 3b. In the left column, a 1D SRG with 5 μm periodicity is patterned on a contracted hydrogel. In the right column, the same SRG is subsequently erased with UV light, restoring the film to the flat state by re-contracting the previously exposed areas. Both the visible interference patterning and the UV erasure were performed using 0.2 s exposure time and 200 mW cm$^{-2}$ intensity. Notably, the contraction and expansion dynamics continues to evolve beyond the illumination period, indicating that the material response lags behind the light stimulus. In addition, Fig. 3a, b reveals surface features indicative of local hot-spots, which are likely caused by inhomogeneities in the intensity distribution of the laser beam used for the patterning. This demonstrates the potential of hydrogel surfaces as beam-profiling tools. Increasing the illumination intensity above 200 mW cm$^{-2}$ did not significantly accelerate the contraction response, suggesting that for the studied hydrogel composition and thickness, the dynamics is limited by intrinsic material relaxation and diffusion processes (Fig. S18). Higher intensities also likely lead to the photothermal effect as compression continues slowly with longer exposure times, which is supported by slightly increased temperature (Fig. S19). Once the light is removed the film shows moderate back-relaxation as the temperature begins to decrease closer towards LCST.

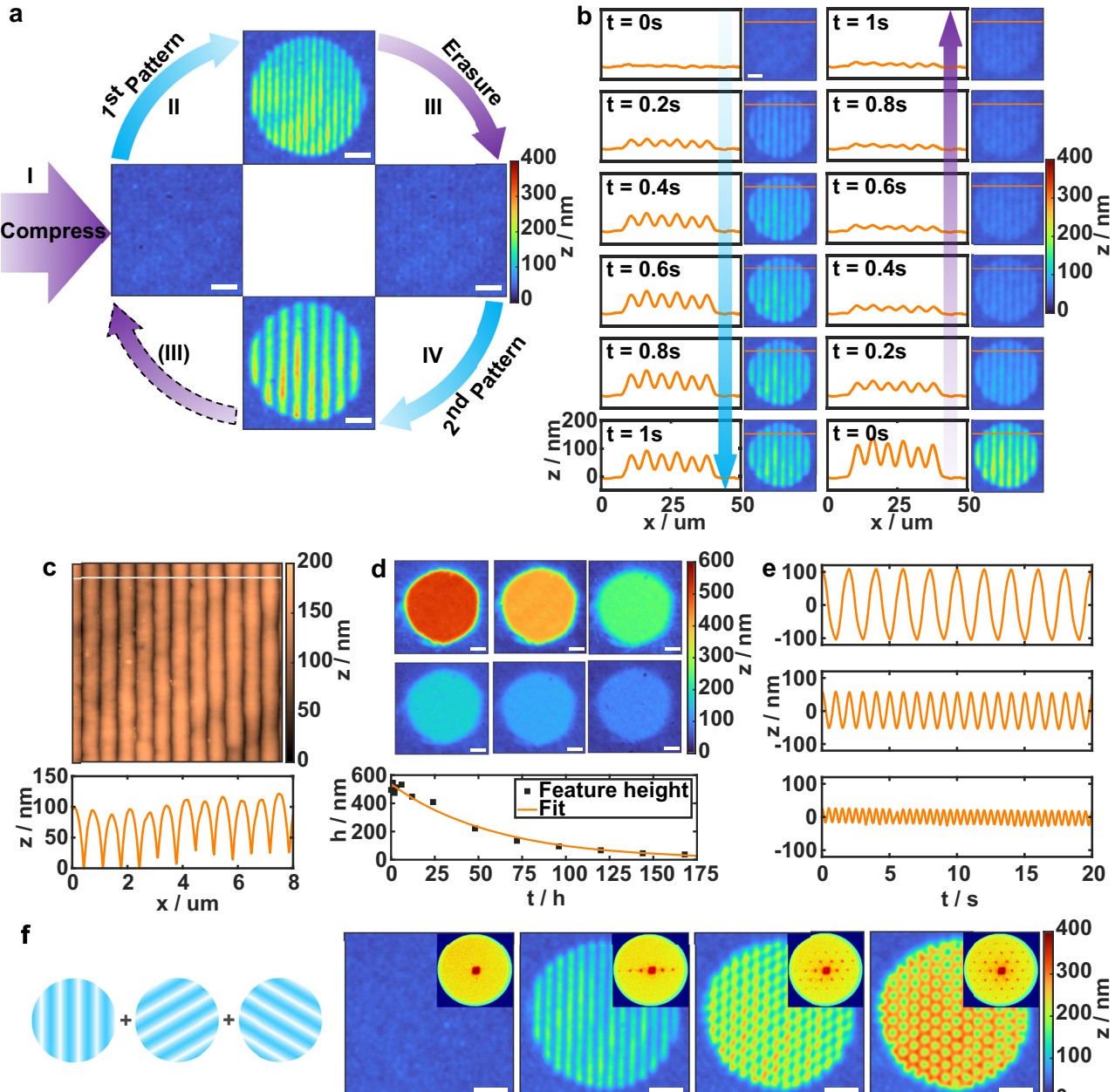

**Fig. 3 | Surface structure pattering dynamics, stability and reconfigurability.** **a** Patterning–erasure cycles of two surface relief gratings (SRGs) on a hydrogel film. The process involves: (I) UV-induced compression (365 nm), (II) visible-light patterning (488 nm) of a 3 μm periodicity SRG, (III) UV-erasure, and (IV) subsequent patterning of a 5 μm periodicity SRG. **b** SRG formation and erasure dynamics under 488 nm interference pattern and 365 nm uniform illumination, respectively. Profiles were extracted along the indicated lines (intensities: 200 mW cm⁻²; Exposure times: 0.2 s). **c** AFM image and surface profile of an 800 nm periodicity grating, patterned with 488 nm (100 mW cm⁻², 2 s). **d** Relaxation of a cylindrical surface feature, showing topography (top) and feature height decay with an exponential fit (bottom) over time. **e** Hydrogel film thickness modulation at frequencies of 0.5 Hz (top), 1 Hz (middle), and 2 Hz (bottom). Irradiation intensities: 50 mW cm⁻² (365 nm) and 200 mW cm⁻² (490 nm). **f** Hexagonal 2D SRG patterning via sequential 1D exposures (3 μm periodicity) with 60° rotation between exposures. Conditions: 488 nm, 200 mW cm⁻², 300 ms. All experiments were conducted in 100 mg mL⁻¹ αCD aqueous solution. DHM images were acquired from Ag-coated samples; scale bars: 10 μm.

The spatial resolution of surface features can be controlled down to a sub-micron scale, as shown in Fig. 3c for a 1D SRG with 800 nm period (488 nm, 100 mW cm⁻², 2 s). Irradiation dosage can be used to control the amplitude and profile of SRGs, with higher dosage causing the amplitude to increase and valleys to grow in (Fig. S20). This is likely due to enhanced Z–E conversion, and hence re-complexation, under the areas of constructive interference. The amplitude of SRGs additionally depends on the spatial period of the pattern and with the current system, the amplitude increases from

100 nm to 300 nm as the SRG period increases from 800 nm (Fig. 3c) to 2 μm (Fig. S20).

To assess the stability of the patterned surface features, we created a cylindrical protrusion and monitored its height relative to the surrounding hydrogel surface under ambient conditions in the dark (Fig. 3d). The cylinder height gradually decreased from an initial value of ca. 500 nm, due to thermal Z–E relaxation of the AZO pendants, which causes the surrounding areas to elevate around the protrusion due to re-complexation. An exponential fit yielded an estimated time

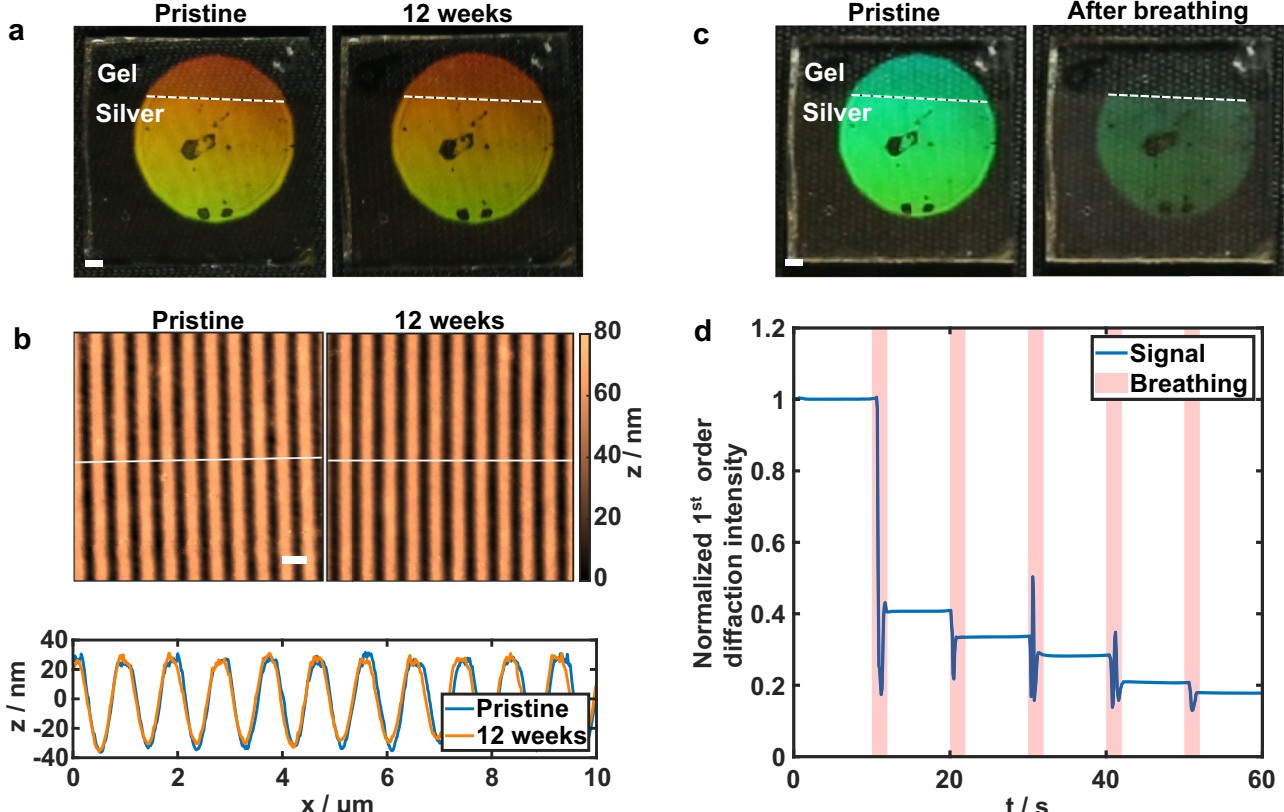

**Fig. 4 | Humidity dependent stability of dried surface structures. a** Long-term stability of dry hydrogel films. Photographs show diffraction from an SRG immediately after patterning (left) and after 12 weeks of storage (right) in ambient conditions. The increased brightness in the lower section is because of the Ag-coating. **b** AFM images and corresponding height profiles of the uncoated SRG region, confirming structural preservation over time. **c** Humidity-triggered erasure. Photographs of a dry hydrogel SRG before (left) and after (right) nine breathing cycles, demonstrating rapid loss of diffraction efficiency. **d** Typical relaxation response of normalized 1st order diffraction intensity from an SRG without Ag coating as a function of time during breathing. Scale bars: (**a, c**) 1 mm, (**b**) 1 μm.

constant of $59 \pm 11$ h (95% confidence bounds), with feature height remaining above 400 nm for the first 24 h. This behavior is in good agreement with $70 \pm 5$ h (95% confidence bounds) time constant of thermal relaxation of $Z$-AZO in the hydrogel (Fig. S21). The relaxation behavior of the cylinder features some irregularities within the first 6 h, which is likely due to a combination of material relaxation following the initial exposure and $E$-AZO population not reaching 100 % under the exposed area, thus not expanding fully. This leads to a slight overall expansion of the hydrogel, including the protruded cylinder, during the early phase of relaxation.

To further benchmark the contraction and expansion dynamics, a hydrogel film was driven into stable oscillation through repeated switching between UV and visible illumination (uniform illumination, no interference pattern) at controlled frequencies. The film thickness modulation under three different switching frequencies, using 365 nm light at 50 mW cm⁻² and 490 nm light at 200 mW cm⁻², is shown in Fig. 3e. At 0.5 Hz frequency, the amplitude of the thickness modulation is 200 nm. This amplitude decreases to 100 nm at 1 Hz, and further to 40 nm at 2 Hz. These results demonstrate rapid and reversible response of the hydrogel, with actuation frequencies spanning over important physiological processes such as the human cardiac cycle (around 1.2 Hz at rest)[53].

The features that can be patterned onto the hydrogel surface are not limited to single-pattern exposures, and more complex structures, such as 2D SRGs or arbitrary patterns, can be generated via multiple exposures or drawn by scanning the beam on the sample. For example, Fig. 3f shows a 2D SRG created by three subsequent exposures of 1D interference pattern, each rotated by 60°, resulting in honeycomb-like structure. The insets show the corresponding Fourier transforms of the SRGs, representing the far-field interference patterns after each exposure and demonstrating the possibilities for diffractive optical elements. We also showed drawings of user-defined patterns, such as logos and a Turing pattern (Fig. S22, Movie S1), highlighting the versatility of the proposed methodology in dynamic, high-resolution surface structuring.

Interestingly, when the hydrogel film is dried after patterning, the patterned SRGs remain fully stable under ambient conditions and normal room lighting. Figure 4a, b show photographs of a hydrogel film with a large-area SRG (patterned with a two-beam interference setup, see Methods) and corresponding AFM images and profiles from plain hydrogel section immediately after patterning and drying and after 12 weeks, revealing virtually no change in appearance or SRG morphology. However, upon exposure to moisture through direct wetting or simply breathing onto the sample, the SRG is erased. Photographs of the hydrogel before and after multiple breathing cycles are shown in Fig. 4c. Notably, the color from the SRG on plain hydrogel is effectively erased while the area covered with Ag-coating shows slighter fading, suggesting that the Ag-layer hinders SRG pattern relaxation via moisture exposure. In addition, Fig. 4d shows a typical behavior of normalized reflected 1st order diffraction signal from SRG on plain hydrogel during breathing. The diffraction intensity decreases rapidly during the first breathing, followed by smaller steps with following breathing cycles. The extent of SRG relaxation relates to the parameters of breathing, i.e., duration.

## Controlled directional movement of the surface features

The controlled dynamics of the hydrogel contraction and expansion allows for live-shaping of surface features via simultaneous exposure

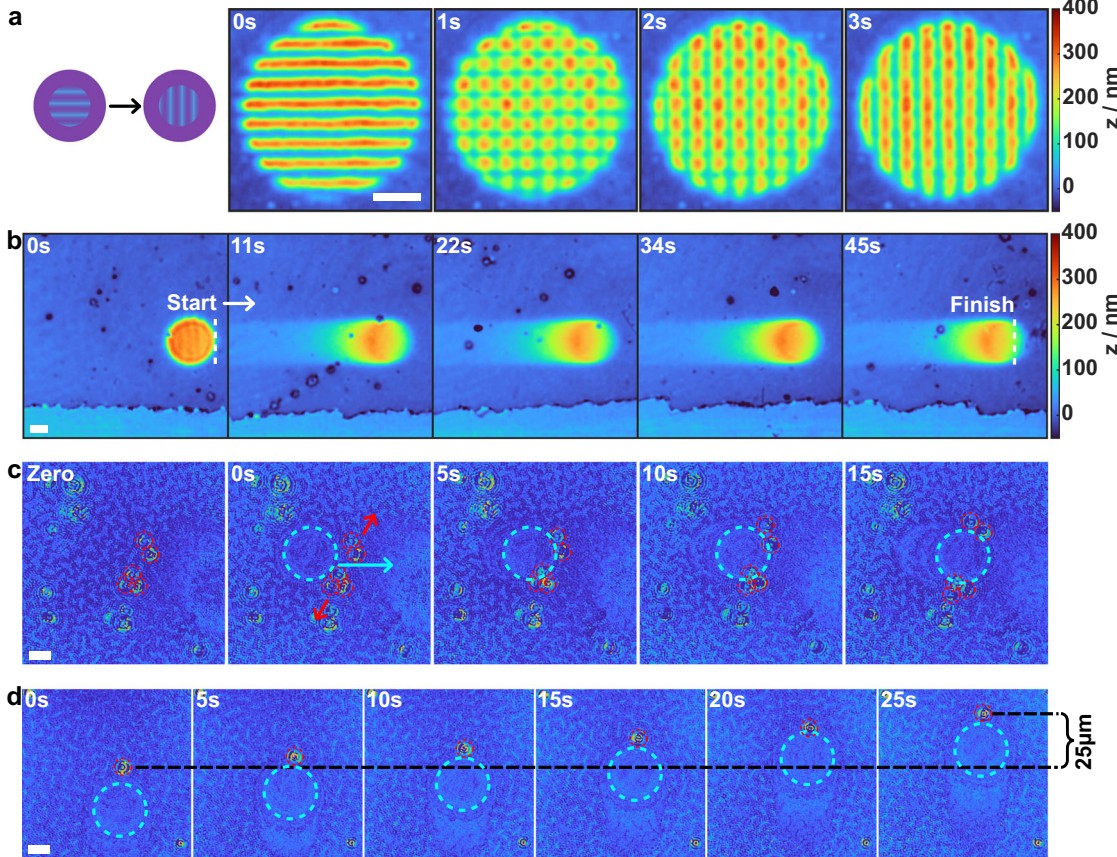

**Fig. 5 | Real-time morphing of surface structures with object-transporting capability. a** Dynamic reorientation of a 1D SRG, rotating the grating vector by 90° from vertical to horizontal. Irradiation: 50 mW cm⁻² at 365 nm; 400 mW cm⁻² at 488 nm. **b** Lateral motion of a cylindrical protrusion, traveling 440 μm across the hydrogel surface in 45 s. Irradiation: 20 mW cm⁻² at 365 nm; 100 mW cm⁻² at 488 nm. **c** Particle separation via translating cylindrical protrusion. Irradiation: 100 mW cm⁻² at 365 nm; 200 mW cm⁻² at 488 nm. **d** Transportation of a single particle using cylindrical protrusion under the same irradiation conditions as in **c**. In **c, d**, the translating particles and protrusions are highlighted in red and light blue, respectively. Data in **a, b** were acquired from Ag-coated samples; data in **c, d** were obtained without coating. All experiments were performed in 100 mg mL⁻¹ αCD solution. scale bars: 10 μm.

with UV and visible light. Under these conditions, surface structures are driven by light-controlled dynamic equilibrium between complexed and free AZOs, enabling real-time modulation of feature shape and prominence by adjusting the spatial pattern and intensity ratio of the two light beams. To demonstrate this, we exposed the sample with a large-area UV beam and used a smaller visible light pattern within the exposed region to realize live features. As an example, we reoriented an SRG with a vertical grating vector to a horizontal one, as shown in time-lapse in Fig. 5a and Movie S2. As continuous UV illumination constantly drives the AZOs from the *E*- to *Z*-isomer, leading to contraction of the hydrogel, the previously expanded regions are erased as soon as the visible exposure area is shifted.

The dual exposure technique also enables the creation of directional movement of surface features. An illustrative example is presented in Fig. 5b, where a cylindrical protrusion is actively translated across the surface of the sample. The presented time-lapse frames are stitched back-to-back to visualize the trajectory of the moving protrusion. Again, UV light effectively erases the protrusion on the exposed path, while visible light advances the expansion front, guiding the motion direction. In this demonstration, the protrusion travels 440 μm in 45 s, corresponding to an average speed of 10 μm s⁻¹. Similarly, continuous movement can be combined with interference lithography to create, for example, traveling waves at a fixed position by shifting the phase between the two interfering laser beams (Movie S3).

The moving surface features can also be harnessed for object manipulation. For this we fabricated a thicker, 1.7 μm (dry thickness, SI,

Fig. S23) hydrogel film for transporting 5 μm borosilicate glass spheres. The increased film thickness introduces surface waviness and creasing[54], but allows higher protrusions to be generated, which is favorable for particle manipulation. In Fig. 5c a group of five glass spheres is separated into two subgroups by guiding a protrusion from left into the centre of the particle cluster. This increases the separation distance of the particles from 6 μm to 20 μm. Another example is shown in Fig. 5d, where a single particle is transported by physically pushing it with a protrusion, covering a distance of 25 μm in 25 s. In addition, we used a traveling wave with a fixed position on the hydrogel surface as a conveyor belt, enabling the transport of multiple particles, as shown in Movie S4

## Shape-morphing of floating hydrogel films

Inspired by dynamic biological surfaces like chameleon and cephalopod skin, we developed a method to impart adaptive functionality to free-standing hydrogel films. We imprinted a passive SRG onto the hydrogel film via hot embossing prior to crosslinking (see Methods, Fig. S24). A section of the film was then detached and set to float on 100 mg mL⁻¹ αCD solution. Upon illumination, the film undergoes reversible, isotropic contraction and expansion, which modulates the periodicity and amplitude of the SRG. This leads to a shift in the diffraction angle, thereby altering the color perception of the film surface at a specific viewing angle, resembling the structural coloration of chameleon skin. As shown in Fig. 6a, under 2 s exposure with 100 mW cm⁻² intensity, a color change from green to blue is obtained when the film contracts by 22 % in area. The demonstrated 2 s exposure time corresponds to a

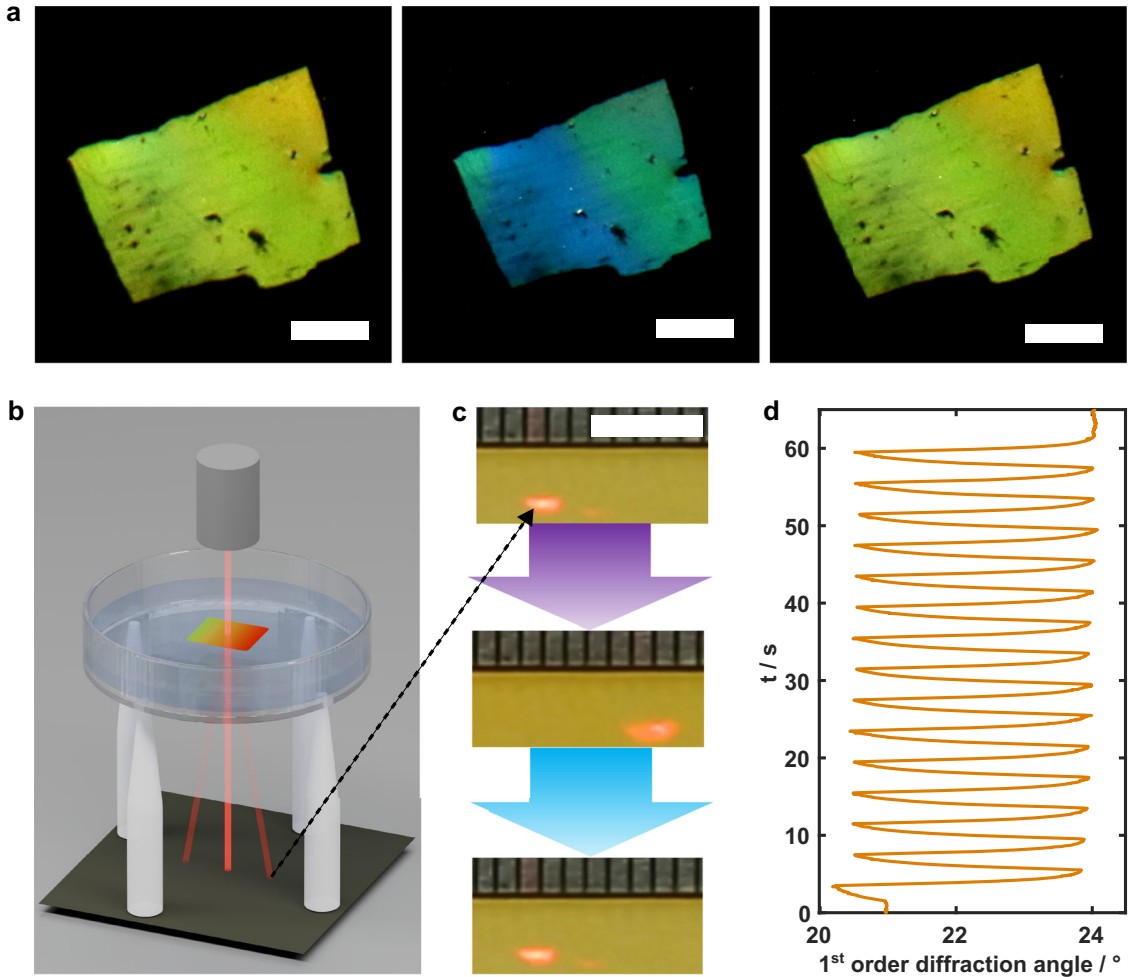

**Fig. 6 | Free-standing hydrogel films for dynamic camouflage and light-controlled photonic devices. a** Dynamic structural color tuning of a free-standing hydrogel film during a 0.25 Hz contraction–expansion cycle, observed at a fixed viewing angle. Scale bars: 1 mm. **b** Schematic of programmable laser beam steering. A free-standing film with a pre-inscribed SRG floats on an αCD solution; a laser beam incident from above produces a transmission diffraction pattern. **c** Spatiotemporal control of beam deflection. Snapshots show the first-order diffraction beam position shifting during a single 0.25 Hz actuation cycle. Scale bar: 5 mm. **d** Continuous beam steering performance. Angular displacement of the first-order diffraction spot over 1 minute of cycling at 0.25 Hz. Irradiation intensities: 100 mW cm$^{-2}$ for 365 nm and 490 nm across all experiments.

stimulation frequency of 0.25 Hz, matching the average human respiratory rate and hence mimicking the contraction–expansion cycle of lungs[53] (Movie S5). Beyond visual appearance, the same mechanism can be used for tunable photonic functionalities such as beam steering. The experimental setup in which a laser beam is directed through a patterned, free-standing hydrogel film, and the transmitted first-order diffraction is monitored is illustrated in Fig. 6b. During the cyclic contraction and expansion, the diffraction angle can be shifted periodically by up to 3.5°, as shown in Fig. 6c, d.

## Discussion

We have developed a versatile method for dynamic surface pattern formation on thin films of light-responsive hydrogels, enabling high spatial resolution and fast temporal response. The underlying mechanism relies on local contraction and expansion of the hydrogel, triggered by light-responsive AZO:αCD inclusion complex formation and disruption. The thickness of the hydrogel film can be modulated at frequencies up to 2 Hz, which exceeds the resting cardiac cycle. At full contraction and expansion, the film achieves nearly 90% thickness modulation. Surface relief gratings can be patterned in a fully reconfigurable manner within a few seconds and with sub-micron resolution. Moreover, the surface features can be stabilized by drying the

hydrogel and subsequently erased by exposure to moisture, demonstrating potential use as moisture-sensitive tags. In addition to reversible, sequential patterning and erasure, we demonstrate real-time control of surface features through simultaneous exposure to UV and visible light. This allows dynamic live-shaping of features, such as reorientation of SRGs, generation of moving surface waves, and directional object translocation on the hydrogel surface. Finally, we extended our approach from surface-bound films to free-standing hydrogel films floating on water surface that undergo rapid and reversible actuation. The high level of control and sensitivity achieved with the presented system indicates a promising direction for hydrogel-based photonic devices, and suggesting potential applications in adaptive optics, dynamic filtering, beam profiling and environmental sensing. In parallel, the proposed platform holds promise for biomedical applications, such as dynamic cell culture platforms that deliver structural and mechanical cues, and actuating systems that model physiological functions like breathing.

## Methods
### Materials
N-isoprpylacrylamide (NIPAm, 99%), Azobisisobutyronitrile (AIBN, 98%, recrystallized from methanol), α-cyclodextrin (αCD, 98%),

3-(trimethoxysilylpropyl) methacrylate (TMSPM, 98%), and diethyl ether (99.8%) were acquired from Merck and 1,4-dioxane (99.5%) from VWR. 4-acrylamidoazobenzene (AZO) and 4-acrylamidobenzophenone (BP) were synthesized inhouse (SI, Figure S1–S2).

## Sample fabrication
Synthesis of PNIPAm-AZO was carried out using thermally initiated free radical polymerization. The precursor mixture was prepared by dissolving NIPAm, AZO, BP, and AIBN in 1,4-dioxane at $100\,mg\,mL^{-1}$ concentration. The fraction of AIBN was kept constant at 2 mol-% while AZO and BP were varied between 0–8 mol-%. The precursor mixture was purged with $N_2$ for 10 min and polymerization carried out at 70 °C for at least 12 h. The product was precipitated and washed with diethyl ether and dried under vacuum for 12 h.

Hydrogel films were fabricated via spin-coating. Microscope glass slides were used as a substrate except for crosslinking kinetics measurements, for which quartz was used due to higher transmittance at 300 nm. Substrates were cut to ca. $1.25 \times 1.25\,cm^2$ squares and cleaned with acetone and IPA using a soft cloth. Glass substrates were surface activated with $O_2$-plasma for 5 min (PDC-002, Harrick Plasma) and subsequently treated with TMSPM by immersing substrates in 1 v-% IPA solution for 1 h. Finally, the substrates were rinsed with IPA and dried with $N_2$-flow. Spin-coated films were applied from $60\,mg\,mL^{-1}$ 1,4-dioxane solution using a dynamic dispensing method with 2000 rpm for 1 min. The samples were annealed for 10 min at 150 °C prior to photocrosslinking, which was conducted with 300 nm LED (Thorlabs, M300L4 mounted LED) at $3\,mW\,cm^{-2}$ and exposure time from 30-300 min depending on the composition. The required exposure times were determined by monitoring the decrease of BP absorption at around 300 nm over time (SI, Fig. S25). After crosslinking, the samples were annealed at 150 °C for 10 min. When required, for example in DHM experiments, an Ag-coating was sputtered (Q150R ES, Quorum Technologies) on a dry hydrogel film with default settings corresponding to 2 nm thickness.

Thicker films for particle transportation (Fig. S23) were fabricated as described earlier except for using spin-coating solution concentration of $150\,mg\,mL^{-1}$, resulting in a dry thickness of 1.7 µm. Crosslinking time of 540 min was used based on the fixed dosage per thickness from determined crosslinking kinetics measurements. The 5 µm borosilicate glass spheres were treated with $O_2$-plasma for 5 min (PDC-002, Harrick Plasma) prior to their application onto hydrogel films from aqueous suspension.

Free-standing films were fabricated as earlier described up to the spin coating step, after which an SRG was hot-embossed on the sample by setting the sample on a hotplate at 180 °C for 10min and then gently pressed with a PDMS mold with an SRG using a glass slide on top for uniform pressure. The PDMS mold was prepared by curing PDMS on top of a commercial grating. After pressing, the sample was removed from the hotplate, and the mold was peeled off after cooling to room temperature. The sample was then crosslinked as described earlier. For further experiments, a piece of hydrogel with pre-inscribed SRG was removed by peeling with a razor blade and set to float on top of $100\,mg\,mL^{-1}$ $\alpha$CD solution. The sample was illuminated with 365 nm and 490 nm LED (pE-4000, CoolLed) for contraction and expansion. A red diode laser (CPS635F, Thorlabs) beam directed on the sample from above was used to track the transmitted first-order diffraction beam. The diffracted beam was traced with ImageJ TrackMate[55].

## Nuclear magnetic resonance (NMR)
NMR spectra were measured with a 500 MHz JEOL ECZR 500 (126 MHz for $^{13}C$) at room temperature. Chemical shifts ($\delta$) are reported in units of parts per million relative to tetramethylsilane. All $^1H$ NMR spectra were referenced to proton signals of residual non-deuterated solvents and $^{13}C$ spectra were referenced to carbon signals of the solvent. Coupling constants (J) are given in Hertz (Hz). MestReNova (version 15.0.1) was used for the evaluation of the NMR data. The following description was used for the multiplicity of the signal: s = singlet, d = doublet, t = triplet, m = multiplet, b = broad. Signals in the experimental part of this work are given as shown below:

Chemical shift $\delta$ [ppm] (multiplicity, coupling constant J [Hz], integral, assignment)

## Size-exclusion chromatography
The molar masses were determined with an Agilent 1100 HPLC system connected to Phenomenex Phenogel (5 µm, 5nm, and 100nm) columns and a UV detector at 280nm. THF was used as an eluent with a flow rate of 1.0 mL/ min. Calibration was conducted using toluene and polystyrene standards. The sample and standards were dissolved in eluent at a concentration of 2 mg/ml and filtered using 0.2 µm syringe filters before the SEC analysis.

## Spectroscopy
UV-Vis experiments were conducted with commercial spectrometer (Cary 60 UV-Vis, Agilent) with a custom-made measurement chamber allowing in-situ illumination. Solutions were measured using a temperature-controlled cuvette holder (qpod 2e, Quantum Northwest). Films were measured using an optical filter holder (M-PPL17, Newport) as is or with a custom-made glass cell allowing sample immersion in fluid. Sample illumination was done with 365 nm and 490 nm LED (pE-4000, CoolLed).

## Hydrogel film morphing
The contraction–expansion of hydrogel films was realized with 365 nm and 490 nm LED (pE-4000, CoolLed) coupled to AFM or DHM. AFM imaging (Dimension Icon, Bruker) was done with SCANASYST-AIR (Bruker, imaging in air), SCANASYST-FLUID (Bruker, imaging in fluid), and SAA-SPH-1UM (Bruker, mechanical imaging) tips. In-situ illumination was achieved by coupling the LED light source to the AFM using a commercial coupler (Photoconductive module, Bruker), which allowed sample exposure through a transparent window from below. Intensities in AFM experiments were estimated from Lorentzian beam intensity profile (beam profiler LBP2-H2-VIS2, Newport) via numerical integration over a 10% peak of total power, resulting in $I_{peak} = 30\,mW\,cm^{-2}$ for 365 nm and $I_{peak} = 70\,mW\,cm^{-2}$ for 490 nm. Digital holographic microscopy (DHM®-R2100, Lyncée Tec) in reflection mode was used for real-time monitoring of the hydrogel surface. Samples were illuminated from the side with 365 nm and 490 nm LED for exposing the whole area under examination. In addition, surface features were created by using 365 nm LED for contracting and p-polarized (perpendicular to grating vector) 488 nm laser (Genesis CX 488–2000 SLM, Coherent) for patterning, with the exception of SRG pattern orientations differing from horizontal grating vector, in which case the polarization direction is s (parallel to vertical grating vector) or combination of s and p polarization vector components. Laser beams were controlled with custom-built illumination setup coupled to DHM[52]. Large-area SRG patterning for dry hydrogel was done with 365 nm LED (pE-4000, CoolLed) and 488 nm laser on two-beam interference setup. Sample was inserted in a custom-made glass cell filled with $100\,mg\,mL^{-1}$ $\alpha$CD solution, compressed with 365 nm LED and exposed to interfering laser beams. Subsequently, the sample was lifted from the cell so that excess solution was drained from the film surface, resulting in predominantly dry film.

## Data availability

The experimental data are available from the corresponding author on request. Source Data file has been deposited in Figshare under accession code https://doi.org/10.6084/m9.figshare.31212109.

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

## Acknowledgements
A.P. acknowledges the financial support from the Research Council of Finland (Flagship programme PREIN, No. 320165; Center of Excellence LIBER, No. 346107), and the European Research Council (Consolidator Grant project MULTIMODAL, No. 101045223). M.P. is grateful to the Finnish Cultural Foundation for financial support. This work made use of the Tampere Microscopy Center facilities at Tampere University. We thank Suvi Lehtimäki for technical assistance with the AFM measurements.

## Author contributions
A.P. and M.P. conceived the project. M.P. fabricated samples, designed and carried out experiments, and analyzed the data. H.M. synthesized AZO and BP monomers and executed and characterized NMR studies. A.B. automated laser drawing setup combined with DHM. A.P. supervised the project. M.P. and A.P. wrote the manuscript. All authors commented on the manuscript.

## Competing interests
M.P. and A.P. are named as inventors on patent application filed based on these findings by Tampere university. M.P. and A.P. declare no other competing interests. The other authors declare no competing interests.

## Additional information

 **Patent applicant:** TAMPERE UNIVERSITY FOUNDATION SR

