## [Transparent Peer Review file · Nature Communications]

Live-shaping of hydrogel thin films with light

Corresponding Author: Professor Arri Priimagi

Version 0:

Reviewer comments:

Reviewer #1

(Remarks to the Author)

Summary of content

Authors show the synthesis of a copolymer that can be selectively swelled and deswelled using light. The swelling properties are controlled by the alpha-cyclodextrin /Azobenzene host/guest interaction. Authors show that the surfaces of hydrogels can be reversibly switched at very low response times and high resolutions. They show free standing films and surface attached films. They show the writing and reversibly erasing of different structures as well as the transport of particles on surfaces.

Overall Assessment

This work is interesting and it is nice to see such a real-time response from a material - and in a reversible form, that is rare. Authors have seemingly put a lot of work into experiments and analyses.

However, the presentation and writing makes it hard to follow this paper. Figure captions do not explain but rather state what was done. The presentation is often confusing and could be simplified to make it easier for the reader. Additionally, the introduction is much too short and does not adequately represent the state of the art. I recommend major revision with specified comments below.

Comments for revision

All figures: captions are not according to standard. All figures should have an overall caption of what is shown and not start with a) ... also captions should explain what is shown (with similar wording as in the text) and not simply state a method used to create the data.

Introduction: Please include recent work, especially on the alphy Cyclodextrine host/guest interactions and hydrogels. Right now the introduction merely provides summarized sentences with generally stating some expamples and then providing a number of citations. There should be an in-depth presentation of the state of the art of the field of hydrogels with re-writable patterns and for this material system in particular. It should also comment on the gels with the fastest response and the state of the art concerning resolution of structured hydrogels.

Figures: 1 strongly recommend to remove the gray background in the figures

Figure 1 caption: please explain in your captions what can be seen. For instance "swelling comparison" is not a good explanation. You display arrows in the image, please explain stating something like "a free standing hydrogel film can expanded upon swelling to all three dimension, while a surface attached film only swells in z-direction" and include an x/y/z

Figure 1 a: should contain a legend with brown: hydrogel greenish: surface

Figure 1b: the CAD renderings at the bottom are not clear and do not help for understanding of what is shown. If these will be kept in the image, please label them with information to make them more clear.

Figure 1 d: the word "compress" is not even mentioned in the caption. Please make your captions match your figures. It is mentioned very late in the text for the first time.

Figure 2 a: I also find the explanation of this figure confusing and hard to understand. "illuminated" from below – does this mean the illumination is a static line pattern at the end of the film? Or is this illumination moving with the AFM measurement tip to create what is shown in figure 2b? 2a seems to be a thickness evaluation of a film on a substrate – as stated in the text, but the caption A schematic for both Figure 2a inset and figure 2b on how these patterns were actually created would have been easier to understand. Also in line with this confusion: Line 131: what do you mean by "scanning across a scratch?" Is a

scratch the illuminated “ditch/valley”?

Figure 2a it is confusing, that this is a schematic but contains actual numbers that do not match the final result. The inset suggests a step of 1000 nm depth, but the data suggests 500 nm. Also, if this is a schematic, it would make more sense to write “distance in nm” instead of “scan direction” as this corresponds to the distance.

Figure 2b if this is a schematic as well, please provide the schematic for how you made this structure – consecutive illuminations of the light or simultaneous illumination of a line pattern with a mask

Figure 2c: h/hwet, what is this? Please explain it in the caption

Line 201 Silver coating: please put in the Mat/met how this sputtering was achieved with the gel, in what state, what conditions. Please also comment on the stability of this silver film, what happens upon scratching, will a scratch appear of will this silver layer be “skin-like” and move away as a skin layer? How thick is this layer? Was this silver coating applied to all of the gels in the experiments? Please state this in the figure captions always.

Line 206 how much slower is the light-response with the silver-coated gel, please provide data and information.

Line 223: where is this lag between illumination and expansion shown? How long is this lag and are all images of patterns left for a certain time to get an equilibrium state? Please comment on this and if all patterns have to rest for some time please include it in the mat/met

Figure 3 a/b: please explain why the illuminated patterns are not uniform: the peaks show distinctly higher areas (which were avoided in the profiles by using a cut at the upper field of the image)

Lines 251 and following: how fast do the structures disappear if not kept in the dark?

Line 273 “direct laser writing” is a term for 2-photon-polymerization – I find it confusing, what is meant by direct laser writing here?

Figure 4 c: what could be the reason for the slight shifts in some of the lines but not all of them?

Videos: the videos show many bumps and speckles on the gel surfaces. Please provide an image of a pristine gel surface and the quality of the surface and an evaluation of the roughness of the gels.

Line 344 hot embossed pattern: Please provide an image visualization of this pattern. please provide image data of the films after hot embossing a static/passive SRG. Please comment on to what extent this pattern will be kept and if the swelling will be isotropic and provide data for this.

Supplementary

Figure crossrefs are not correct in the Supplementary. Figure S1 is an NMR and does not show UV/VIS Data. The mistake continues for other figures. Please correct this.

If you provide equations like ES1, please provide the calculation or all the values used for an example to make it easier for the reader to understand what was done/calculated

Mat/Met

Line 410 can authors please comment on the necessity of the annealing step at 150 °C and what this helps with/causes?, likewise for the next annealing step (line 415)

Can authors please comment on the long illumination times for making the gel films? What about the absorption maximum of AZO relative to BP? Will the two be interfering with each other?

Reviewer #2

(Remarks to the Author)

This manuscript by Priimagi and co-workers describes a system of light-responsive hydrogel films with swelling that can be reconfigured on a time-scale of approximately one second, allowing for ‘real time’ shaping of surface topography. All of the basic ingredients of the system are already established (see Refs. 36 – 38), but the authors manage to combine a quite large light-induced change in swelling (close to two-fold in thickness) with a rapid response time, and a microscope system with patterned illumination that allows them to demonstrate optically-rewritable surfaces that far exceed prior reports in terms of spatio-temporal control. Thus, I support publication and have only relatively minor questions/comments below.

Have the authors studied the thickness-dependence of swelling kinetics at all? This could be a way to more clearly establish that they are in the mass-transfer limited regime (at least at high light intensities.)

I am puzzling over the kinetics data in Figure S10, and wonder if the authors can provide more interpretation. First, I presume there is no photo-thermal effect, so if the light is switched off, the samples do not recover? Second, the approach to a limiting curve at high intensities and short times suggests they likely are in the mass-transport limited regime, but if so, photoisomerization is relatively quick and one would think (with simple photo-switching kinetics) that the photostationary state (PSS) is reached within less than ~ 1 s, and thus it is not clear how the slow transient at high intensities would be reflective of reaching the PSS. Also, given the very long relaxation time, I do not think that the Z-isomer content at the PSS should be substantially intensity dependent.

In Video S4, particle transportation is clearly non-uniform, with at least one isolated particle and a collection of particles getting ‘stuck’ in place. Can the authors comment on why this may be?

I do not see a description for Video S5 in the SI.

Reviewer #3

(Remarks to the Author)

[Editorial Note: Please see end of file]

Version 1:

Reviewer comments:

Reviewer #1

(Remarks to the Author)

Authors have addressed all previously raised issues. I recommend publication of the work in the current form.

Reviewer #2

(Remarks to the Author)

In most respects, I am happy with the authors' responses and revisions. However, I do have additional concerns related to swelling kinetics.

In the newly added figure S14, the authors show that swelling kinetics do not depend on thickness of the hydrogel film. First, they should specify the light intensity used for this experiment. Second, this is exactly the opposite of what would be expected if the swelling kinetics were mass-transport limited—the classical expectation for a mass-transfer limited process is that the characteristic timescale for swelling should scale as thickness squared.

The data in Figure S18 (here the authors should add the film thickness) suggest that at 50 and 100 mW/cm² (where the curves differ), the process is not mass-transport limited, but rather set by photochemical switching kinetics, whereas by 200 – 300 mW/cm² (the curves appear to lie on top of each other initially) they may have reached the mass-transport limit.

I don't think that any additional experiments are required, but the authors may want to update their discussion of these results.

Reviewer #3

(Remarks to the Author)

The authors have addressed all my concern. It can be published as it is now.

REVIEWER COMMENTS

Reviewer #1

Summary of content

Authors show the synthesis of a copolymer that can be selectively swelled and deswelled using light. The swelling properties are controlled by the alpha-cyclodextrin /Azobenzene host/guest interaction. Authors show that the surfaces of hydrogels can be reversibly switched at very low response times and high resolutions. They show free standing films and surface attached films. They show the writing and reversibly erasing of different structures as well as the transport of particles on surfaces.

Overall Assessment

This work is interesting and it is nice to see such a real-time response from a material - and in a reversible form, that is rare. Authors have seemingly put a lot of work into experiments and analyses.

However, the presentation and writing makes it hard to follow this paper. Figure captions do not explain but rather state what was done. The presentation is often confusing and could be simplified to make it easier for the reader. Additionally, the introduction is much too short and does not adequately represent the state of the art.

I recommend major revision with specified comments below.

OUR ANSWER: We thank the reviewer for their careful inspection of our work and for the insightful notes regarding our presentation, clarity, and experimental methods. We are particularly encouraged by the reviewer's recognition of the rarity and interest in the real-time reversibility demonstrated in our material.

In light of the reviewer's feedback, we have revised the manuscript to improve readability and contextualize our findings within the broader field. Specifically, we have:

- Expanded the Introduction to provide a more comprehensive overview of the current state-of-the-art.
- Rewritten the Figure Captions to ensure they provide a detailed explanation of the results rather than just a summary of the experiments.
- Refined the overall prose to simplify the presentation and make the narrative easier to follow.

We have addressed all specific comments point-by-point below and are confident that these changes have strengthened the manuscript.

All figures: captions are not according to standard. All figures should have an overall caption of what is shown and not start with a) ... also captions should explain what is shown (with similar wording as in the text) and not simply state a method used to create the data.

OUR ANSWER: We have updated the figure captions and included short descriptive titles.

Introduction: Please include recent work, especially on the alpha Cyclodextrine host/guest interactions and hydrogels. Right now the introduction merely provides summarized sentences with generally stating some examples and then providing a number of citations. There should be an in-depth presentation of the state of the art of the field of hydrogels with re-writable patterns and for this material system in particular. It should also comment on the gels with the fastest response and the state of the art concerning resolution of structured hydrogels.

OUR ANSWER: We agree that the introduction lacked content on light responsive hydrogels, especially detailed discussion on those utilizing α CD-azobenzene complexation. We have expanded this section and added references closely related to the current work. Here is the modified section:

“These systems can, in general, enable dynamic crosslinking, thereby significantly improving mechanical properties and self-healing ability³¹⁻³⁴. When such inclusion complexes are made light-responsive through the incorporation of a photoswitchable guest, their properties can be modulated remotely with high spatiotemporal control^{35,36}. An extensively studied example is provided by inclusion complexes between azobenzene photoswitches and cyclodextrins. In these systems, host-guest complexation can selectively favour one isomer over the other, resulting in pronounced hydrophilicity contrast between the two states.

The light-controllable host-guest complexation has been widely implemented, ranging from control over supramolecular gelation³⁷ and self-assembly³⁸ to macroscopic deformations in soft robotic systems³⁹, microactuators⁴⁰, and artificial muscles^{41,42}. Recently, this principle was exploited to create reconfigurable micropatterns on hydrogel films through the integration of azobenzene: α -cyclodextrin complexes. Photoisomerization of azobenzene allows for localized control over film swelling and contraction, yielding surface patterns with spatial resolution down to 40 μ m in less than a minute. Importantly, these patterns can be erased and rewritten by photoisomerization of azobenzene, enabling fully reversible architecture that acts as a versatile micromolding platform for complex surface features.”

Figures: 1 strongly recommend to remove the gray background in the figures Figure 1 caption: please explain in your captions what can be seen. For instance “swelling comparison” is not a good explanation. You display arrows in the image, please explain stating something like “a free standing hydrogel film can expanded upon swelling to all three dimension, while a surface attached film only swells in z-direction” and include an x/y/z

Figure 1 d: the word “compress” is not even mentioned in the caption. Please make your captions match your figures. It is mentioned very late in the text for the first time.

Figure 1 a: should contain a legend with brown: hydrogel greenish: surface ?

Figure 1b: the CAD renderings at the bottom are not clear and do not help for understanding of what is shown. If these will be kept in the image, please label them with information to make them more clear.

OUR ANSWER: We have carefully revised the figure caption and added more in-depth description to match the schematics. We have replaced the arrows with xyz-axis and added legend to differentiate between hydrogel and substrate in Fig. 1a. We added describing labels to Fig. 1b to clarify fabrication step schematics.

Figure 2 a: I also find the explanation of this figure confusing and hard to understand. “illuminated” from below – does this mean the illumination is a static line pattern at the end of the film? Or is this illumination moving with the AFM measurement tip to create what is shown in figure 2b? 2a seems to be a thickness evaluation of a film on a substrate – as stated in the text, but the caption A schematic for both Figure 2a inset and figure 2b on how these patterns were actually created would have been easier to understand. Also in line with this confusion: Line 131: what do you mean by “scanning across a scratch?” Is a scratch the illuminated “ditch/valley”?

OUR ANSWER: We have revised the schematics in Fig. 2a and added xyz-axis, as well as shading effect to AFM tip to clarify the scan direction. In the experimental setup, light beam and sample are stationary and the AFM scanner moves in y-direction. This results in a step-profile are shown in the inset – which additionally presents the difference in film thickness under different conditions. We agree that the phrase “scanning across a scratch” is misleading. This has been replaced by “...scanning across a step profile between the substrate and the hydrogel film for thickness determination.”

Figure 2a it is confusing, that this is a schematic but contains actual numbers that do not match the final result. The inset suggests a step of 1000 nm depth, but the data suggests 500 nm. Also, if this is a schematic, it would make more sense to write “distance in nm” instead of “scan direction” as this corresponds to the distance.

OUR ANSWER: We acknowledge the issue with Fig. 2a containing both schematic and actual data. We have revised the wording in figure caption of the inset as “***Inset: typical film thickness profiles of hydrogel film containing 4 mol-% AZO and 2 mol-% BP in air, water, and α CD solution***” to emphasize this being data and not schematics. In addition, the data in inset does match closely with the data in Fig. 2d, both showing film thickness of ca. 1000-1100 nm.

Figure 2b if this is a schematic as well, please provide the schematic for how you made this structure – consecutive illuminations of the light or simultaneous illumination of a line pattern with a mask

OUR ANSWER: This is a complete AFM image with a purpose to show how the film thickness data was extracted. We have added lightbulb-illustrations in Fig.2 b to show the lighting conditions in different sections.

Figure 2c: h/h_{wet} , what is this? Please explain it in the caption

OUR ANSWER: We realize this abbreviation lacked proper explanation. The term “wet” was changed to “water”. The abbreviation h/h_{water} refers to film thickness h relative to film thickness in water h_{water} .

Line 201 Silver coating: please put in the Mat/met how this sputtering was achieved with the gel, in what state, what conditions. Please also comment on the stability of this silver film, what happens upon scratching, will a scratch appear or will this silver layer be “skin-like” and move away as a skin layer? How thick is this layer? Was this silver coating applied to all of the gels in the experiments? Please state this in the figure captions always.

OUR ANSWER: We acknowledge that the description on silver coating preparation was not extensive and lacked instrumentation information. This has been added to the methods section as follows:

“When required, for example in DHM experiments, silver coating was sputtered (Q150R, Quorum Technologies) on a dry hydrogel film with default settings corresponding to 2 nm thickness.”

We have also added data on silver layer thickness and changes after immersion (Fig. S15, see below), and notions on experimental data whether or not silver-coated sample was used:

Fig. 3: *“All DHM images were acquired from samples with Ag coating.”*

Fig. 4e: *“Typical relaxation response of normalized 1st order diffraction intensity from an SRG without Ag coating as a function of time during breathing.”*

Fig. 5: *“Data in (a, b) acquired with Ag coating, (c, d) without.”*

Figure S15: AFM images and corresponding average profiles of **a** pristine and **b** 10 min submerged silver layer on glass substrate, sputtered with default settings for 2 nm. Digital holographic microscope (DHM) **c** intensity and **d** phase images over boundary between silver and hydrogel surface. Scale bars: 10 μm.

Line 206 how much slower is the light-response with the silver-coated gel, please provide data and information.

OUR ANSWER: Unfortunately, with current means we are not able to reliably characterise this. Of the two methods available, DHM cannot resolve hydrogel surface without silver coating and AFM can only provide illumination from below the sample. Thus, we have based the hypothesis on additional attenuation caused by silver coating, which reduces the optical power reaching the hydrogel compared to uncoated sample.

Line 223: where is this lag between illumination and expansion shown? How long is this lag and are all images of patterns left for a certain time to get an equilibrium state? Please comment on this and if all patterns have to rest for some time please include it in the mat/met

OUR ANSWER: The mentioned lag is present in Fig. 3b time series, where the exposure is stated to be 0.2s and the surface structure is observed to grow after this time point. The timescale to fully equilibrated state can be estimated from Fig. S18. With minimal

photothermal effect while illuminating with 50 mW cm^{-1} the film contraction slows down significantly after 10s. We did not leave the samples to equilibrate for any particular time, however, static patterns were imaged after ca. 10s after no significant changes were observed.

Figure 3 a/b: please explain why the illuminated patterns are not uniform: the peaks show distinctly higher areas (which were avoided in the profiles by using a cut at the upper field of the image)

OUR ANSWER: This phenomenon arises from non-uniformity of the laser beams used for the pattern inscription. We added a following mention of this to the text: *“In addition, Fig. 3a-b reveal surface features indicative of local hot-spots, which are likely caused by inhomogeneities in the intensity distribution of the laser beam used for the patterning. This demonstrates the potential of hydrogel surfaces as beam-profiling tools.”*

Lines 251 and following: how fast do the structures disappear if not kept in the dark?

OUR ANSWER: This depends solely on the lighting conditions, i.e., intensity and spectrum. Higher intensity at blue or UV region of the spectrum leads to faster relaxation/erasure.

Line 273 “direct laser writing” is a term for 2-photon-polymerization – I find it confusing, what is meant by direct laser writing here?

OUR ANSWER: We appreciate pointing out this confusion. This is now replaced with “drawing” with laser beam.

Figure 4 c: what could be the reason for the sight shifts in some of the lines but not all of them?

OUR ANSWER: The minor changes in period are probably because of the combination of different imaging spot and slight variation in two-beam interference angle because of substrate and fluid cell glass window curvature. We have added a following reference to the main text regarding the fabrication of this SRG pattern: *“...hydrogel film with a large-area SRG (patterned with a two-beam interference setup, see Methods)”*.

Videos: the videos show many bumps and speckles on the gel surfaces. Please provide an image of a pristine gel surface and the quality of the surface and an evaluation of the roughness of the gels.

OUR ANSWER: We have added data in SI from roughness of pristine and used samples. The bumps and speckles are likely due to agglomeration of silver layer and/or aggregation of α CD on hydrogel surface after multiple long experiments (Fig. S17).

Figure S17: AFM images of 4% Azo 2% BP hydrogel as **a** pristine, and **b,c** after multiple long experiments without (**b**) and with (**c**) the silver coating with corresponding mean surface roughness parameters S_a .

Line 344 hot embossed pattern: Please provide an image visualization of this pattern. please provide image data of the films after hot embossing a static/passive SRG. Please comment on to what extent this pattern will be kept and if the swelling will be isotropic and provide data for this.

OUR ANSWER: We have added AFM images of the SRG on master mold, PDMS replica mold, and hydrogel into SI, Fig. S24 (see below).

Figure S24: AFM images from **a** master SRG, **b** PDMS replica, and **c** dry hydrogel before delamination, with **d** corresponding surface profiles extracted from red lines.

Swelling is expected to cause expansion of the SRG, resulting in increases in both the period and amplitude, while simultaneously reducing the refractive index and, consequently, the diffraction efficiency. The light response is observed to be isotropic in orthogonal directions, corresponding to an approximately 11% contraction. Further systematic investigation of changes in the SRG or refractive index was not feasible with the current experimental techniques.

Line 410 can authors please comment on the necessity of the annealing step at 150 °C and what this helps with/causes?, likewise for the next annealing step (line 415) Can authors please comment on the long illumination times for making the gel films? What about the absorption maximum of AZO relative to BP? Will the two be interfering with each other?

OUR ANSWER: The primary purpose of the first annealing step is to ensure complete solvent evaporation. The second annealing step is intended to thermally relax the azobenzene units after crosslinking via 300 nm UV exposure; however, this step is not critical for patterning, as the initial UV illumination typically induces compression. The relatively long illumination time required for crosslinking at 300 nm is solely limited by the available light source. In addition, a

slight overlap between the BP and azobenzene absorption peaks (see below) partially slows the crosslinking process, although it does not prevent it.

Figure R1. Spectral changes during crosslinking of 2 mol-% AZO, 2 mol-% BP hydrogel film, corresponding to data presented in Fig. S25a. Blue curve: 0 min; Green curve: 180 min.

Supplementary

Figure crossrefs are not correct in the Supplementary. Figure S1 is an NMR and does not show UV/VIS Data. The mistake continues for other figures. Please correct this.

OUR ANSWER: We realise this mistake has made the inspection of the work unnecessary demanding and apologize for that. The cross references are now updated.

If you provide equations like ES1, please provide the calculation or all the values used for an example to make it easier for the reader to understand what was done/calculated
Mat/Met

OUR ANSWER: As suggested, we have included the values used in calculations.

Reviewer #2:

This manuscript by Priimagi and co-workers describes a system of light-responsive hydrogel films with swelling that can be reconfigured on a time-scale of approximately one second, allowing for ‘real time’ shaping of surface topography. All of the basic ingredients of the system are already established (see Refs. 36 – 38), but the authors manage to combine a quite large light-induced change in swelling (close to two-fold in thickness) with a rapid response time, and a microscope system with patterned illumination that allows them to demonstrate optically-rewritable surfaces that far exceed prior reports in terms of spatio-temporal control. Thus, I support publication and have only relatively minor questions/comments below.

OUR ANSWER: We thank the Reviewer for their positive assessment and for recognizing that our system far exceeds prior reports in terms of spatio-temporal control. We are pleased that the Reviewer appreciated the combination of the large light-induced swelling response with the rapid, real-time reconfiguration. We also appreciate the insightful minor questions and comments, which have helped us further clarify the technical nuances of our work. We have addressed each point below and made the corresponding updates to the manuscript.

Have the authors studied the thickness-dependence of swelling kinetics at all? This could be a way to more clearly establish that they are in the mass-transfer limited regime (at least at high light intensities.)

OUR ANSWER: Thank you for the insightful comment. In response, we investigated the relationship between film thickness and swelling kinetics. No significant differences were observed among dry film thicknesses of 60, 120, and 400 nm (Fig. S14). This suggests that the process is mass-transfer limited, at least under the typical light intensities used in this work.

Figure S14: Light-responsive contraction-expansion of hydrogel films with 4 mol-% AZO and 2 mol-% BP and dry thicknesses of 60, 120 and 400 nm.

I am puzzling over the kinetics data in Figure S10, and wonder if the authors can provide more interpretation. First, I presume there is no photo-thermal effect, so if the light is switched off, the samples do not recover? Second, the approach to a limiting curve at high intensities and short times suggests they likely are in the mass-transport limited regime, but if so, photoisomerization is relatively quick and one would think (with simple photo-switching kinetics) that the photostationary state (PSS) is reached within less than ~ 1 s, and thus it is not clear how the slow transient at high intensities would be reflective of reaching the PSS. Also, given the very long relaxation time, I do not think that the Z-isomer content at the PSS should be substantially intensity dependent.

OUR ANSWER: Thank you for pointing out a feature that we initially overlooked. Indeed, the film thicknesses show a gradual recovery toward a common value after the light source is switched off (Fig. S17). This suggests the presence of a photothermal component at higher intensities, leading to a greater extent of film contraction by raising the film temperature further above the LCST.

To investigate this further, we attempted to probe the film temperature during illumination by imaging the sample from below using a thermal camera. The sample was spin-coated onto a thin microscope coverslip, after which the film was partially removed from the outer edges with a razor blade, leaving a square hydrogel film at the center of the substrate. This geometry was used to distinguish film heating from potential substrate heating. The sample was then covered with a water droplet to mimic the standard experimental conditions. Despite the indirect nature of the temperature measurement, an increase of approximately 0.8 °C was observed after illuminating the sample with 365 nm UV light at 200 mW cm⁻² for 1 min (Fig. S19). In addition, heating was clearly localized to the central region of the substrate covered by the hydrogel, supporting the presence of a minor photothermal effect under the experimental conditions used.

Figure S19: **a** Thermal images of a hydrogel film containing 4 mol-% AZO and 2 mol-% BP spin-coated onto a microscope coverslip. Images were acquired from below (substrate side) with a water droplet on top while illuminating

*the sample from above. Temperatures represent averages over the square regions indicated in the images. **b** Temperature profiles extracted from the images in (a) along the rectangular regions indicated. Scale bar: 1 mm.*

In Video S4, particle transportation is clearly non-uniform, with at least one isolated particle and a collection of particles getting 'stuck' in place. Can the authors comment on why this may be?

OUR ANSWER: We attribute this behavior to adhesion between the particles and the hydrogel surface. Although this was not investigated in further detail, we observed that, without O₂-plasma treatment, it was difficult to induce particle motion. We therefore hypothesize that the hydrogel surface remains relatively hydrophobic and interacts sufficiently strongly with untreated particles to immobilize them, irrespective of surface manipulation. In the present example, partial particle adhesion is likely due to imperfect surface treatment or aging of the hydrophilic modification.

I do not see a description for Video S5 in the SI.

OUR ANSWER: We have added the missing description to Video S5.

Reviewer #3:

In this manuscript, the authors have prepared a light-responsive hydrogel thin film platform capable of rapid, reconfigurable surface modulation with sub-micron spatial resolution and actuation frequencies up to 2 Hz. Based on the photoswitchable host-guest interaction, reversible expansion-contraction is realized in response to patterned illumination. Dual-wavelength control results in the generation of dynamic, migrating surface features, which can transport micro-objects in real time. Moreover, surface patterns can be stabilized by drying and erased by humidity, offering a route to rewritable sensor tags. This is an interesting study and can be suitable for publication in Nature Communications. However, several minor points should be addressed.

OUR ANSWER: We thank the Reviewer for their positive evaluation and for highlighting the key strengths of our platform, including the sub-micron spatial resolution and the high actuation frequencies. We are pleased that the Reviewer found our demonstrations of particle transport and rewritable sensor tags interesting and suitable for Nature Communications. We have carefully addressed the minor points raised by the Reviewer, which have been instrumental in improving the technical rigor and the description of the device's functional performance. Our point-by-point responses and the corresponding revisions in the manuscript are detailed below.

1. The ¹H NMR spectrum, molecular weight, and polydispersity of poly(NIPAm-co-AZO-co-BP) should be provided in this work.

OUR ANSWER: We thank the reviewer for pointing this out, and acknowledge that this information was overlooked in the initial submission. We have added the requested

information in SI (Fig. S3 – S5). In addition, the term referring to yet uncrosslinked material was changed from “polymer” to “oligomer” based on revised experiments.

Figure S3: ¹H-NMR spectra of AZO and BP monomers and oligomer (4 mol-% AZO, 2 mol-% BP). Arrow depicts peak in oligomer resulting from AZO monomer, while others are result from both monomers.

Figure S4: $^1\text{H-NMR}$ spectra of 3 oligomer samples (4 mol-% AZO, 2 mol-% BP).

Figure S5: SEC profiles of intermediate oligomers with 2 mol-% AZO and 1, 2, and 4 mol-% BP.

2. Please present the direct evidences for the binding behavior between poly(NIPAm-co-AZO-co-BP) and αCD , such as $^1\text{H-NMR}$ titrations.

OUR ANSWER: In response to the comment raised, we have further studied the binding between poly(NIPAm-co-AZO-co-BP) and αCD with $^1\text{H-NMR}$. Both titration and photoswitching experiments were performed, and the results support the original findings from the UV-Vis titration (Figs. S7–S8). In addition, the UV-Vis titration was repeated with longer time averaging to reduce spectral noise.

Figure S7: **a** 1H-NMR spectra of titration with $2,56 \text{ mg mL}^{-1}$ poly(PNIPAm-co-AZO) (0.4 mM of AZO) and [CD] 0 – 51,3 mM increasing from bottom to top. **b** [CD] vs. $\delta_{8.1 \text{ ppm}}$ peak integral in **a** with fit corresponding ES5 & ES3.

Figure S8: 1H-NMR spectra of 2.56 mg mL^{-1} poly(PNIPAm-co-AZO) (0.4 mM of AZO) with 51.3 mM αCD as relaxed (before illumination, E-rich), and after 365 nm illumination (Z-rich).

3. The reversible photo-isomerization of AZO in poly(NIPAm-co-AZO-co-BP) without/with the addition of αCD can be determined by UV-vis absorption spectroscopy. Please refer it.

OUR ANSWER: We have added the isomerization spectra to the Supporting Information (Fig. S8). No significant differences are observed between the cases with and without αCD .

4. How about the mechanical properties of the free-standing hydrogel films?

OUR ANSWER: Unfortunately, free-standing films are too fragile to be accurately characterized using the available instrumentation. However, we measured the modulus change of a surface-attached film during illumination (Fig. S13). An approximately 2.5-fold increase in modulus was observed during hydrogel film contraction, from 270 to 690 kPa. These values are reasonable for hydrogels; however, the absolute values should be considered indicative, as the mechanical analysis is highly sensitive to AFM tip- and cantilever-related parameters.

Figure S13: AFM mechanical imaging of hydrogel film with 4 mol-% AZO and 2 mol-%BP. **a** Height image zeroed to compressed thickness. **b** Elastic modulus image. **c** Zeroed height and modulus profiles, averaged between 200 – 310 nm on x-axis in **a** and **b**.

REVIEWER COMMENTS

Reviewer #2:

In most respects, I am happy with the authors' responses and revisions. However, I do have additional concerns related to swelling kinetics.

In the newly added figure S14, the authors show that swelling kinetics do not depend on thickness of the hydrogel film. First, they should specify the light intensity used for this experiment. Second, this is exactly the opposite of what would be expected if the swelling kinetics were mass-transport limited—the classical expectation for a mass-transfer limited process is that the characteristic timescale for swelling should scale as thickness squared.

The data in Figure S18 (here the authors should add the film thickness) suggest that at 50 and 100 mW/cm² (where the curves differ), the process is not mass-transport limited, but rather set by photochemical switching kinetics, whereas by 200 – 300 mW/cm² (the curves appear to lie on top of each other initially) they may have reached the mass-transport limit.

I don't think that any additional experiments are required, but the authors may want to update their discussion of these results.

OUR ANSWER: We thank the Reviewer for their careful and detailed assessment of the supporting data related to swelling kinetics. We acknowledge that the results of thickness dependence in added Fig. S14 were overlooked and the reasoning was inconsistent. The light intensity in AFM experiments was estimated to be around 30 mW cm⁻² for 365 nm, i.e., not sufficient to push the kinetics om diffusion limited region according to data in Fig. S18. We have therefore made the following changes to the interpretation of these results:

Thickness dependence of light-induced expansion/contraction

The effect of film thickness on the light responsive expansion/contraction was studied by comparing 3 samples with varying nominal dry thicknesses of 60 nm, 100 nm and 400 nm. Samples were imaged with AFM as dry, in water and in 100 mg mL⁻¹ αCD solution. Fig. S14 shows the relative response to film thickness in water (h/h_{water}) when illuminated with $I = 30 \text{ mW cm}^{-2}$ for 365 nm and $I = 70 \text{ mW cm}^{-2}$ for 490 nm.

As no significant differences are present in the rate of contraction and expansion of hydrogel films, the process is assumed to be limited by light intensity and thus mass-transport limit is not reached.

This is in line with the results shown in Fig. S18, where increasing intensity eventually reaches limit after which the contraction response does not change anymore.

Figure S14: Light-responsive contraction-expansion of hydrogel films with 4 mol-% AZO and 2 mol-% BP and dry thicknesses of 60, 120 and 400 nm. Irradiation parameters: 30 mW cm⁻² at 365 nm and 70 mW cm⁻² at 490 nm.

In addition, we have added the light intensity estimations used in AFM experiments to the main text as follows:

“The intensities in all AFM measurements were estimated as 30 mW cm⁻¹ for 365 nm and 70 mW cm⁻¹ for 490 nm (see Methods).” (under **Light-Induced Contraction and Expansion**)

“Intensities in AFM experiments were estimated from Lorentzian beam intensity profile (beam profiler LBP2-H2-VIS2, Newport) via numerical integration over a 10% peak of total power, resulting in $I_{\text{peak}}=30 \text{ mW cm}^{-2}$ for 365 nm and $I_{\text{peak}} = 70 \text{ mW cm}^{-2}$ for 490 nm.” (under **Methods**)

In this manuscript, the authors have prepared a light-responsive hydrogel thin film platform capable of rapid, reconfigurable surface modulation with sub-micron spatial resolution and actuation frequencies up to 2 Hz. Based on the photoswitchable host–guest interaction, reversible expansion–contraction is realized in response to patterned illumination. Dual-wavelength control results in the generation of dynamic, migrating surface features, which can transport micro-objects in real time. Moreover, surface patterns can be stabilized by drying and erased by humidity, offering a route to rewritable sensor tags. This is an interesting study and can be suitable for publication in *Nature Communications*. However, several minor points should be addressed.

1. The ^1H NMR spectrum, molecular weight, and polydispersity of poly(NIPAm-co-AZO-co-BP) should be provided in this work.
2. Please present the direct evidences for the binding behavior between poly(NIPAm-co-AZO-co-BP) and αCD , such as ^1H NMR titrations.
3. The reversible photo-isomerization of AZO in poly(NIPAm-co-AZO-co-BP) without/with the addition of αCD can be determined by UV-vis absorption spectroscopy. Please refer it.
4. How about the mechanical properties of the free-standing hydrogel films?